# A Custom-Tailored Multichannel Pressure Monitoring System Designed for Experimental Surgical Model of Abdominal Compartment Syndrome

**DOI:** 10.3390/s24020524

**Published:** 2024-01-15

**Authors:** Zoltan Attila Godo, Katalin Peto, Klaudia Balog, Adam Deak, Erzsebet Vanyolos, Laszlo Adam Fazekas, Zsolt Szentkereszty, Norbert Nemeth

**Affiliations:** 1Department of Information Technology, Faculty of Informatics, University of Debrecen, Kassai Str. 26, H-4028 Debrecen, Hungary; godo.zoltan@inf.unideb.hu; 2Department of Operative Techniques and Surgical Research, Faculty of Medicine, University of Debrecen, Moricz Zsigmond Str. 22, H-4032 Debrecen, Hungary; deak.adam@med.unideb.hu (A.D.); vanyolos@med.unideb.hu (E.V.); fazekas.laszlo@med.unideb.hu (L.A.F.); nemeth@med.unideb.hu (N.N.); 3Department of Surgery, Faculty of Medicine, University of Debrecen, Moricz Zsigmond Str. 22, H-4032 Debrecen, Hungary; balog.klaudia@med.unideb.hu (K.B.); szentkereszty.zsolt@med.unideb.hu (Z.S.)

**Keywords:** multichannel pressure monitoring system, abdominal compartment syndrome, experimental model

## Abstract

In experimental medicine, a wide variety of sensory measurements are used. One of these is real-time precision pressure measurement. For comparative studies of the complex pathophysiology and surgical management of abdominal compartment syndrome, a multichannel pressure measurement system is essential. An important aspect is that this multichannel pressure measurement system should be able to monitor the pressure conditions in different tissue layers, and compartments, under different settings. We created a 12-channel positive–negative sensor system for simultaneous detection of pressure conditions in the abdominal cavity, the intestines, and the circulatory system. The same pressure sensor was used with different measurement ranges. In this paper, we describe the device and major experiences, advantages, and disadvantages. The sensory systems are capable of real-time, variable frequency sampling and data collection. It is also important to note that the pressure measurement system should be able to measure pressure with high sensitivity, independently of the filling medium (gas, liquid). The multichannel pressure measurement system we developed was well suited for abdominal compartment syndrome experiments and provided data for optimizing the method of negative pressure wound management. The system is also suitable for direct blood pressure measurement, making it appropriate for use in additional experimental surgical models.

## 1. Introduction

Abdominal compartment syndrome (ACS) is a dangerous and challenging condition in clinical practice [1,2,3,4,5]. According to the definition of the World Society of the Abdominal Compartment Syndrome (WSACS), intra-abdominal hypertension (IAH) is a condition where the pressure in the abdominal cavity is 12 mmHg or above [6]. The abdominal perfusion pressure (APP) is a well-accepted marker for characterization of intra-abdominal pressure circumstances. APP is the difference of mean arterial pressure (MAP) and IAP. The fourth stadium of IAP is abdominal compartment syndrome, where the intra-abdominal pressure is 25 mmHg or more accompanied by newly developed organ failure [6,7,8]. IAH/ACS is primary if the cause of the pressure elevation is abdominal or pelvic disease, such as intra-abdominal hemorrhage, paralytic ileus, etc. In these cases, surgical or radiological intervention is often necessary. It is secondary when the cause is an extra-abdominal condition. If two or more compartments (thoracic, intracranial, etc.) are affected, it is called poly-compartment syndrome [2,6,7,8].

The pathomechanism of IAH/ACS is not clearly known yet. The elevated IAP affects not only the abdominal organs but also other compartments. Direct compression of the abdominal organs and blood vessels reduces blood supply to the liver, kidneys, pancreas, and intestinal tract, causing failure and damage to other organs [6,7,8,9,10,11]. The treatment of IAH/ACS is usually conservative and semiconservative (percutaneous fluid drainage). Decompression of the gastrointestinal tract, sedation, muscle relaxation, and diuretic or dialysis therapy are the most commonly used conservative treatments. For localized or diffuse abdominal fluid evacuation, percutaneous drainage is recommended. Surgery is only indicated if other treatments have failed. The most accepted surgical treatment is decompressive laparotomy and open abdomen with negative pressure wound therapy (NPWT/VAC) [6,7,8,9,10,11,12].

To understand the correct mechanism of IAH/ACS and to develop effective therapies, animal experiments are needed. For investigation of pathomechanism and different drug treatments, small experimental animal (mice, rats, etc.) models are commonly used. For testing the effects of surgical therapy, large experimental animal (rabbits, pigs, etc.) models are proposed [10,13,14,15,16,17,18].

In clinical practice, monitoring abdominal pressure is based on pressure measurement in intra-abdominal organs such as the bladder, stomach, etc. Bladder pressure—the most commonly used method—correlates with intra-abdominal pressure but does not give information about the pressure in different parts of the abdomen. This question is even more interesting when negative pressure wound therapy is used for intra-abdominal hypertension or abdominal compartment syndrome. Our multi-channel system for continuous pressure measurement is sensitive to pressure changes in different parts of the abdomen. The sensors are sufficiently flexible and can be fixed with stitches or glue to well-defined parts of the abdominal organs (e.g., diaphragmatic surface of the liver). With this system, the effectiveness of the inserted protective layer can be monitored not only in animal experiments but also in human practice.

With the classical methods (e.g., intra-abdominal pressure measurement via urinary bladder catheter) it is impossible to separately test pressure relations on the layers, especially in the case of negative tissue wound therapy. Therefore, for comparative studies of the complex pathophysiology and surgical management of abdominal compartment syndrome, multichannel pressure monitoring becomes essential. Therefore, our purpose is to fabricate a proper, portable device.

## 2. Materials and Methods—Description of the Device

### 2.1. Sensors

Choosing the right pressure sensor for the task is essential. The most common off-the-shelf pressure sensors operate on three different principles: capacitive, inductive, and piezoresistive [19,20]. However, compared with inductive pressure sensors, piezoelectric pressure sensors are more common [21].

The main constituent element of capacitive sensors is a membrane. The principle of pressure measurement is based on this. The sensor has a membrane that forms a capacitor with another plate in a fixed position. The capacitor capacity changes due to the distance between the two armatures of the capacitor formed in this way, i.e., due to the deformation of the membrane. This capacity value is proportional to the change in pressure, and the distance between the jaws is inversely proportional to the capacity value; they have a high degree of linearity.

The operating principle of inductive pressure sensors is based on Bourdon tube pressure measurement. In terms of its structure, there is a transformer in the sensor whose iron core moves under pressure, changing the degree of inductive coupling between the two coils; this change is proportional to the value of the pressure to be measured.

Piezoresistive pressure sensing is based on the deformation caused by the pressure acting on the diaphragm [19,20], during which the electrical resistance of the deformed material changes. This phenomenon is more noticeable in the case of semiconductor batteries.

We used the MPX5010DP (0–10 kPa ≈ 75 mmHg) and MPX 5050DP (0–50 kPa ≈ 375 mmHg) piezoresistive sensors (Motorola Inc., Denver, CO, USA) for intra-abdominal and blood pressure measurements, respectively.

### 2.2. Analog Noise Filtering

A stable, noise-free power supply is a basic condition for precise measurement. The perfect solution would be a battery with noiseless direct current; however, with proper filtering, the mains power supply is also adequate. A high-capacity capacitor was also installed in order to smooth the signal. The low resistance of the capacitor and the conductor strips together create a low-pass filter, which can eliminate any noise and voltage fluctuations that may occur on the supply voltage. The +5 V supplied by USB did not prove to be stable enough and its voltage also dropped under load; so, we had to discard the USB power supply.

Figure 1 shows the recommended decoupling circuit for interfacing the integrated sensor to the A/D input of a microcontroller. Proper decoupling of the power supply is recommended. Figure 2 shows the stable, noise-free power supply.

Inside the sensor, a fluorosilicone gel isolates the die surface and wire bonds from the environment, while allowing the pressure signal to be transmitted to the sensor diaphragm. Therefore, the membrane is protected and the system can also be filled with water, which is needed later on. For analog-to-digital conversion, the converter IC requires a 2.5 V reference voltage, which is provided by the MAX6225 IC (MAX6225 Low-Noise, Precision, Voltage Reference: Maxim Integrated Products, Inc., San Jose, CA, USA). This provides a high precision constant voltage for analog-to-digital conversion (Figure 3).

The reference voltage is necessary because the value of the signal to be converted must be compared to this voltage during the A/D conversion. This is the basis for the conversion to a digital signal. If the value of this voltage fluctuates, the conversion would become inaccurate. In addition to all this, another element was also placed in the established system, which provides a continuous supply voltage for the clock signal module. The voltage required for the operation of this module is provided by a battery, because during the measurement, which can last several hours, continuous time measurement and the time stamp assigned to the measured values are essential; if the power supply were to stop for any reason during the measurement, the value of the clock signal module would be reset to zero, which is eliminated by the use of the element.

In the intra-abdominal and invasive blood pressure measurement systems, the sensors are placed in two rows, the connectors of which are accessible on the front side of the device (Figure 4).

The reference pressure pipe was also led out next to the pressure measurement pipe. We expected not only negative but also positive pressure in the abdominal cavity. For this purpose, two tubes were inserted at each pressure measurement point. One detected the positive and the other the negative pressure by changing the pressure measuring tubes. So, when the positive pressure is measured at port 1, port 2 is the reference pressure. In case of negative pressure, the 2nd is the measuring point and the 1st connector is the reference pressure. Negative pressure will also appear as a positive value. This can be distinguished by keeping a record of which sensor measures positive and which measures negative pressure. The value of sensors measuring negative pressure must be converted to negative via software.

The measurement system processes the data of a total of 12 pressure sensors, which is performed by a microcontroller. The number of pins on the microcontroller is highly limited and, due to its structure, it can only execute serial instructions, so it would not be able to process the values of all sensors at the same time. It also greatly simplifies the designed circuit if only one signal is processed at a time. Therefore, it is enough to read the value of one selected sensor one after the other in the program—we used a multiplexer for this.

Multiplexers can handle analog or digital signals. If route selection needs to be performed with digital signals, they are usually built from logic gates; in this case, the signals from the sensors are analog. Multiplexers have several data inputs, as well as one data output and, in addition, address inputs for selecting the inputs (Figure 5).

The easiest multiplexer we can obtain has 8 data inputs; to address each input, it must have 3 address inputs (HCF4051 Single 8-channel analog multiplexer/demultiplexer, STMicroelectronics, Geneva, Switzerland). In order for all sensors to be read, 2 multiplexers are needed. Here, for easier handling and addressability later, one of the sensors measuring the positive and the other the negative values are connected to it.

The signals coming from the sensor are received by a multiplexer, whose address inputs are to be addressed by the microcontroller, according to which the analog signal supplied by the pressure sensor is to be connected to the other circuits, thereby performing the route selection. The signal is then fed to the input of an operational amplifier after a voltage divider. Here, the signal is conditioned. This is necessary because it ensures the most accurate possible conversion for the analog-to-digital conversion, as well as the appropriate signal amplification and input impedance matching for the conversion.

A voltage follower circuit was implemented with the operational amplifier, which is a non-inverting amplifier circuit where the gain is of unity. Actually, the output signal is a reproduction of the input signal. The amplifier circuit does not change the phase of the signal and no amplification takes place.

The advantage is that the input of the circuit has a high impedance, while the output signal appears at the output at a low impedance, thereby implementing impedance matching and also functioning as a power amplifier.

### 2.3. A/D Conversion

The circuit provides 4 analog–digital inputs for the central microcontroller using the ADS1115 chip that fits the IIC/I2C bus. It is ideal when a high-resolution signal needs to be processed and the usual 10–12-bit resolution is not enough. The high-precision ADC is provided by the Texas ADS115 chip (AD7715 Sigma-Delta ADC, Analog Devices Inc., Norwood, MA, USA).

Sampling is possible with a maximum of 860 samples/second; the maximum resolution of the circuit is 16 bits. By configuring the chip, it is possible to have 4 independent ADC input channels or to act as 2 differential ADCs. Its internal amplifier can be programmed and a maximum of 16× amplification can be achieved with it; so, the sensitivity can be adjusted in the lower input voltage range. The circuit can be easily adapted to either 3.3 or 5 V systems. The module also has address selection; so, it can be set to 4 different addresses (max. 4 circuits connected, creating max. 16 inputs).

The task of the analog-to-digital converter is to ensure that the value of the digital signal appearing at its output corresponds to the instantaneous value of the analog signal arriving at the input of the converter. For this transformation, a reference voltage is needed, to which the maximum value of the output voltage is compared.

During analog-to-digital conversion, samples must be taken from the analog signal at specific time points. This value must be assigned to a digital number with as little error as possible. The digital value represents the incoming analog signal more precisely; the higher the resolution of the converter, i.e., the number of bits available, depends on how many bits are used for number representation. The AD converter used can convert the analog signal to digital with a 16-bit resolution. This means that between 0 V and 5 V, the 65,535 can distinguish between different voltage levels, i.e., the input voltage change of 0.0763 mV results in a bit change at the output.

The AD7715 IC used is a sigma–delta converter. This type of conversion is performed as follows: The signal arriving at its input is sent to an integrator, and the output of the integrator is monitored by the system through a comparator. By applying a reference voltage to the integrator with the help of a switch, it modifies the output voltage of the integrator so that it converges to zero. The smaller the input voltage, the shorter the reference voltage must be switched on. As a result, fewer bits will appear on the output. The ratio of the analog voltage at its input to the number of individual bits given at its output is the same.

### 2.4. Communication between Components

The devices in the circuit are controlled by the microcontroller. The number of control pins is limited; so, it is necessary to install an expansion IC, thereby increasing the number of control pins. This IC is controlled by the microcontroller with the help of the I2C protocol. According to the control, the two multiplexers are controlled with digital signals, according to which the sensor’s value can be read by the microcontroller. The clock signal IC is also connected to the I2C BUS. This module has a separate, independent power supply. This ensures a stable and reliable clock signal for accurate documentation of the measured data during the measurement.

In addition to I2C communication, the microcontroller can also be connected to a computer via a serial port. By implementing serial communication, the data measured during the measurement can be displayed, processed, and stored on the computer.

### 2.5. The User Interface

The user interface is located on top of the sensor system box. A switch has been placed here, with which the system can be switched on, i.e., it can be placed under power. Furthermore, a 20 × 4 LCD display was placed.

It was necessary to place 3 buttons on the operator interface, which are connected to the microcontroller, and in a specific section of the program, their current value is read out and the program section assigned to the given button is run. By pressing the calibration button, the microcontroller calibrates itself and sets the default values, which will serve as a reference during further operation. In this case, the sensors are not connected to the measuring point, so they can be reset to atmospheric pressure. This is the offset value, which is characteristic of the pressure sensor. The system then performs 10 measurements as quickly as possible. It takes the median of the data and compensates the later measured values with this value. Ten scans give the system a high degree of error tolerance. Erroneous measurement results are shifted to the extreme values, so the middle value is the most relevant. During the data collection, the 5 scanned values prove to be sufficient, the median value of which is also stored. At least 3 out of 5 buffered numbers must be wrong in order for the wrong data to be displayed. Experience has shown that reading 5 numbers provides sufficient error tolerance. Unfortunately, the drift of the set value can also be observed during the measurement. This can be adequately compensated by a warm-up and warm-down of about half an hour.

The third button is used to reset the system time. With the help of this, the beginning of the measurement can be specified. Each time data was saved, an exact time stamp is also saved. In this way, the graph can be reliably evaluated and if there is a break during the recording of the experimental data, the sampling times can still be reliably identified.

The connection of the sensors to the measurement point was solved with a rigid-walled plastic pipe. The soft tube absorbed the pressure, so it was not a suitable solution. The hard-walled tube was filled with physiological saline solution. Thus, a rigid liquid column was created, which perfectly conducted the pressure shock waves. The fluorosilicone gel of the sensors provided protection for the membrane from the liquid.

Figure 6 shows the complete system circuits.

### 2.6. Data Processing

The system is functional even without connecting to a computer. In this case, the microcontroller saves the data. When connected to the computer, the data can be saved in a simple text file. A graphic management program has also been prepared. The program was written in C#. This is suitable for continuous data monitoring. In the middle of the experiment, we can immediately see if incorrect values appear or if any problem arises. The sensors can be turned on and off, so we can go back through the data lines in time. We can enlarge the graph and monitor the correct data reading. We can also save the data in a CSV file so that it can be processed later with any program. Plotting and recording every serial datum from measurements is an essential need; because of the Arduino interface, any universal serial monitor with data plotting capabilities can be used to this machine (e.g., QtSerialMonitor 1.5 by Michal W., open-source software on GitHub Inc., San Francisco, CA, USA), which makes it more cost-effective.

## 3. Results—Applicability

We examined how the analog voltage output of the sensor is related to the measured pressure values. The MPX5050 sensor corresponds to a maximum error of 2.5% between 0 °C and 85 °C; this provides precise data for our experiments (Table 1).

The device was originally designed for ACS study; so, its first application was an experimental model of ACS. In that previously completed study, pressure values were evaluated at different points in the abdominal cavity in a porcine model of experimental abdominal compartment syndrome [16,17]. The ACS was created by implanting a plastic bag into the abdomen, which was filled with body temperature saline solution until an intra-abdominal pressure (IAP) of 30 mmHg was reached. To check the pressure in different parts of the abdominal cavity and in the abdominal organs and blood vessels, sensitive pressure sensors are necessary. After 3 h treatment, negative pressure wound therapy (NPWT) or a Bogota bag was applied. The NPWT group was further divided into −50, −100, and −150 mmHg groups. We wanted to obtain information about how pressure is conducted to the lateral and deeper regions of the abdomen to observe the effectiveness of this NPWT system. A crucial point of negative wound pressure therapy is to maintain a stable and satisfactory level of negative pressure in the deep layers of the abdominal cavity. In the previously completed study, we wanted to know how pressure is transmitted to the lateral and deeper regions of the abdomen [17]. Therefore, six pressure sensors were placed at different points of the abdominal cavity: midline, above the layer; midline, under the layer; laterally, above the layer; laterally, under the layer; midline, among the bowels; and laterally, among the bowels. Figure 7 shows a representative chart with settings of the measurements with six parallel sensors in the ACS model. 

We analyzed the differences between the measurement points and their variation over time. Absolute pressure levels were significantly higher above the protective layer than below. The pressure values were similar in the midline and laterally. Between the intestines, pressure values varied from 0 to −12 mmHg from time to time, which could be attributed to peristalsis. The negative pressure is well conducted to the area under the protective layer in the midline and to the superficial lateral region of the abdominal cavity. The pressure is distributed well between the intestines in the midline and laterally [17].

## 4. Discussion and Conclusions

According to the largest electronic component datasheet search engine, “pressure sensor” returned 19,727 hits (Electronic Components Datasheet Search. https://www.alldatasheet.com, accessed: 9 December 2023); so, the selection is huge. The main aspect of sensor selection is reliability, e.g., contamination tolerance, linearity, low margin of error, and low offset drift. The possibility arose to use wireless pressure sensors. Some research infers the intra-abdominal pressure using an intragastric wireless sensor. The intragastric pressure, validated for estimating IAP, was assessed by an ingestible pressure sensor [22]. However, in this case, a power source, noise filter electronics, and a wireless data transmission circuit should have been connected to the sensor, which would have increased the dimensions. Furthermore, the location of the sensor is not fixed, i.e., it can move. The express purpose of intragastric pressure gauges is to continuously travel through the alimentary canal and then be emptied through the anus. Here, however, it must not be moved from the measurement location. The movement of the intestines, especially as a result of vacuum-assisted closure (VAC), forces the non-fixed sensor to move and would provide information about a false measurement location. Attaching the wireless sensor to tissues—for example, by sewing or gluing—would cause additional irritation in the abdominal cavity, which would distort the effect of artificially induced ACS. Another problem is that we can only record an absolute pressure value in the closed abdominal cavity, not a relative one. It is only possible to reset the external atmospheric pressure to zero before the implantation, after which the change in the intraocular pressure must be detected with an external pressure gauge and the internal measured values compensated with it.

The system we developed has many advantages. The sensors measure in two ranges, 10 and 50 kPa, according to the requirements of the task. The sensitivity of the sensors is extremely high. This is enhanced by noise-filtering electronics and a stabilized power supply. The numerous sensors are connected sequentially to the 16-bit, high-resolution A/D converter by a multiplexer. At this point, a stable reference voltage has been provided. Two sensors were connected to each of the measurement points to detect negative and positive relative pressure. We developed the smallest damping of the pressure wave with a rigid-walled tube filled with liquid. Offset voltage and heating drift were compensated by reading calibration values. Measurement errors were reduced by multiple readings and median calculation. The sampling frequency can be adjusted in software from a hundredth of a second upwards. The data are read and stored by a microcontroller, and then continuously transmitted to a PC. Each sampling is precisely time-stamped for authentic evaluation. On the PC side, we developed graphic processing software in the C# system, where the variable values of the sensors can be graphically monitored. In the event of an error, we can intervene in the measurement process. The data can be saved in the widely known CSV format for later processing.

Multiple readings, A/D conversion, and data processing of the 12 sensors require significant processor time. At a sampling frequency of less than a tenth of a second, sampling does not take place in real time. In this case, the credibility of the graph can be distorted. Higher speeds can only be achieved by using more processors. The distance of the sensors from the sampling location can also be a disadvantage. So, it is the length of the pipe that is used to lead the pressure wave to the device. Damping can occur here as a result from friction with the pipe wall and the flexibility of the pipe. Since the system can be connected to any sterile regular cannulas or tubes used in medicine (with an appropriate connector), there is no direct issue for biocompatibility problems.

We can conclude that for comparative studies of the complex pathophysiology and surgical management of abdominal compartment syndrome, a multichannel pressure measurement system is essential. An important aspect is that this multichannel pressure measurement system should be able to monitor the pressure conditions in different tissue layers, and compartments, under different settings. It is also important that the pressure measurement system should be able to measure pressure with high sensitivity, independently of the filling medium (gas, liquid). The multichannel pressure measurement system we developed was well suited for abdominal compartment syndrome experiments and provided data for optimizing the method of negative pressure wound management. The system is also suitable for direct blood pressure measurement via intravascular catheters, making it suitable for use in additional experimental surgical models.

## Figures and Tables

**Figure 1 sensors-24-00524-f001:**
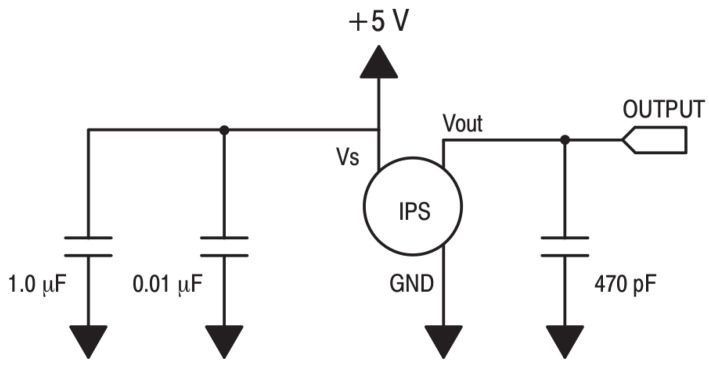
Recommended decoupling circuit.

**Figure 2 sensors-24-00524-f002:**
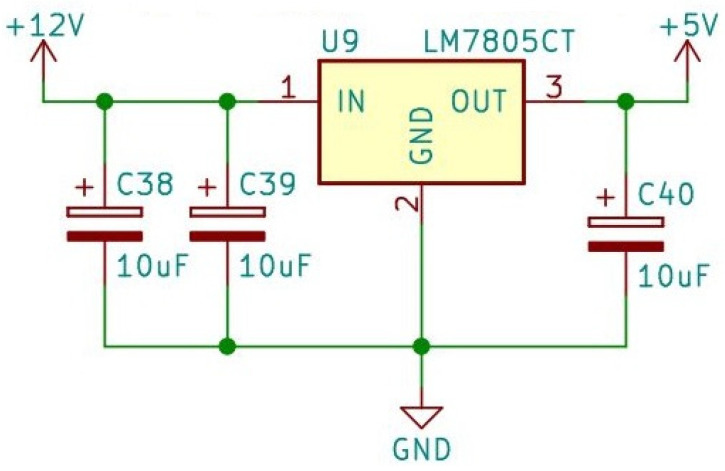
Stable, noise-free power supply.

**Figure 3 sensors-24-00524-f003:**
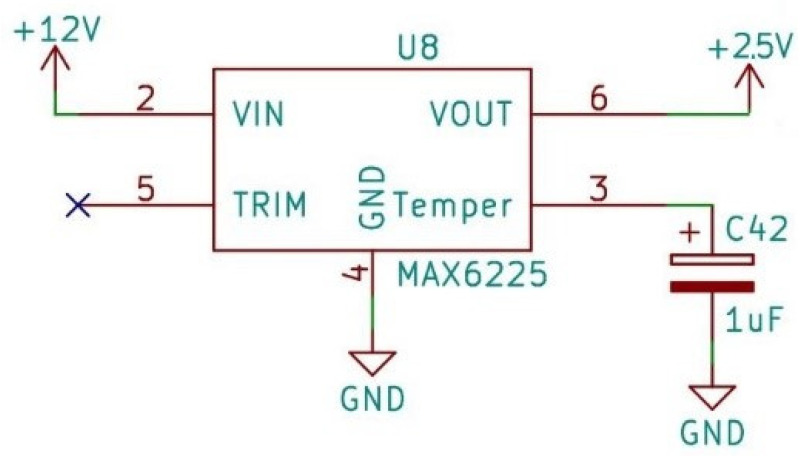
Reference voltage.

**Figure 4 sensors-24-00524-f004:**
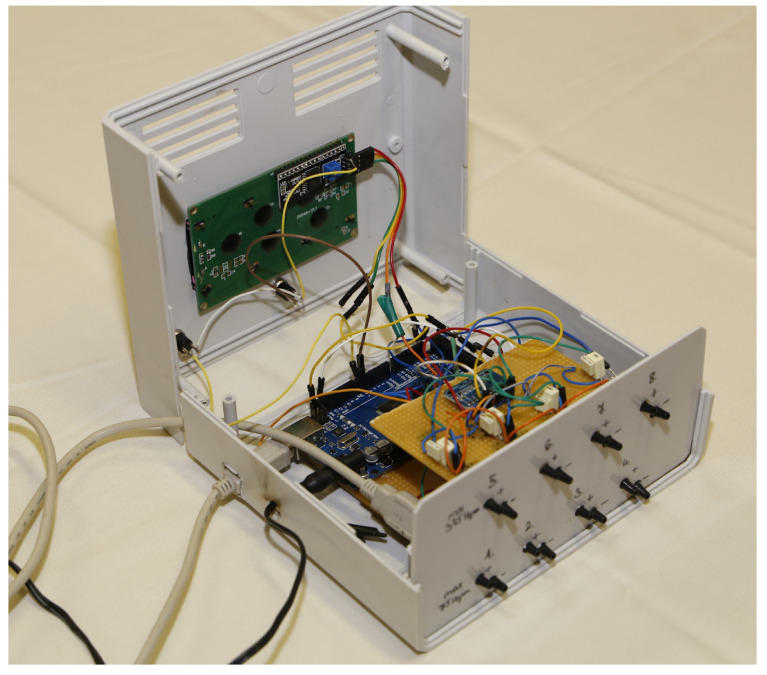
Photo of the intra-abdominal pressure monitoring device.

**Figure 5 sensors-24-00524-f005:**
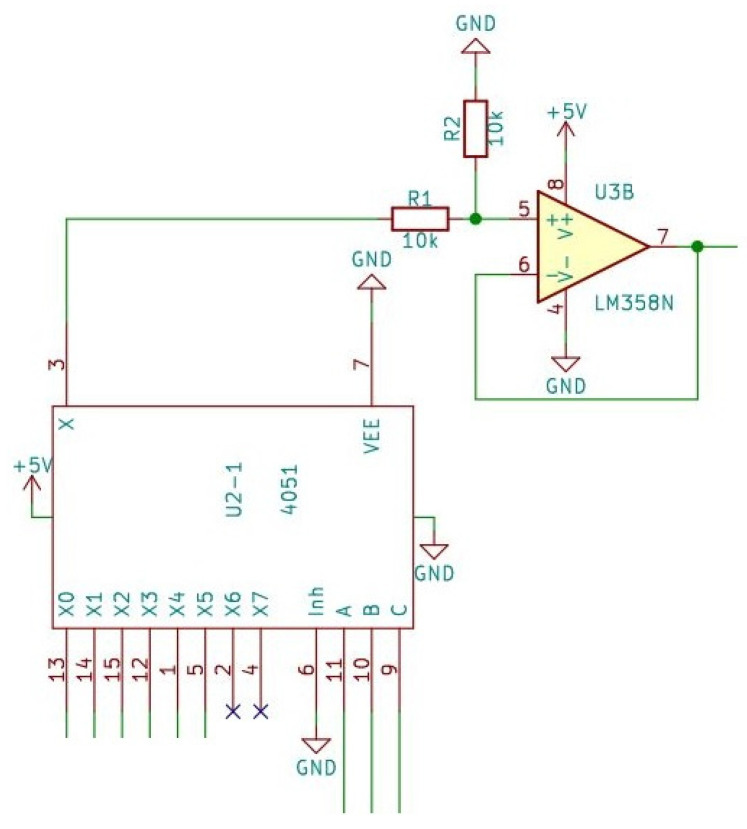
The multiplexer and the voltage follower circuit.

**Figure 6 sensors-24-00524-f006:**
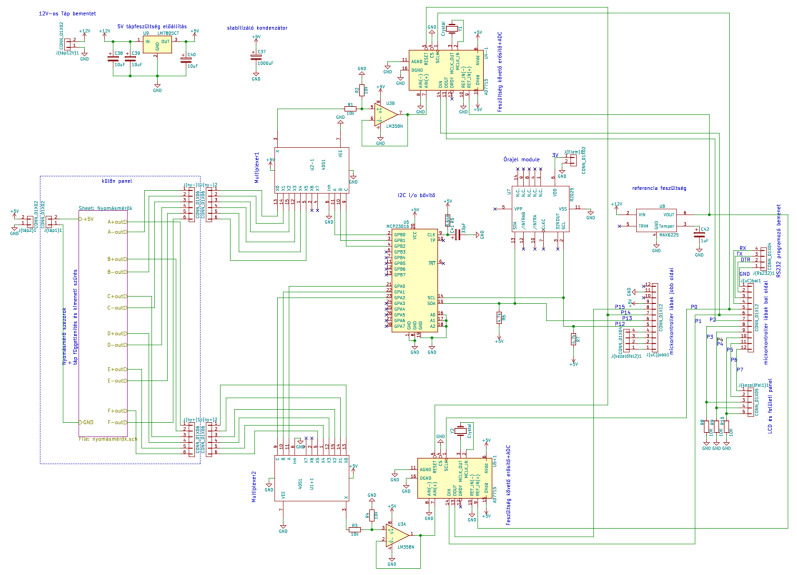
The full system circuits.

**Figure 7 sensors-24-00524-f007:**
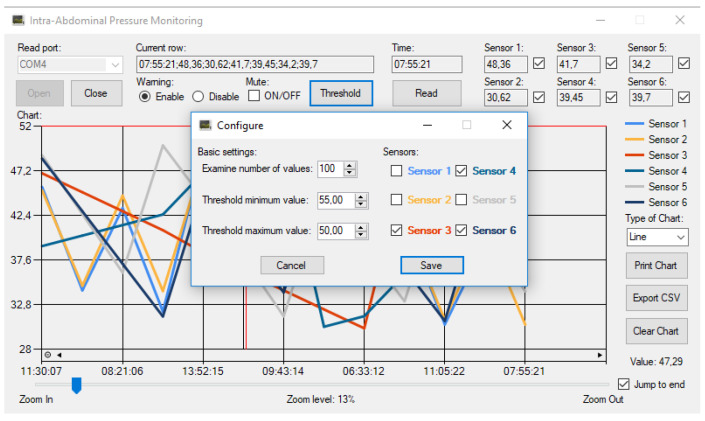
Graphical user interface on a computer—a representative chart in use.

**Table 1 sensors-24-00524-t001:** Relation of the sensor’s analog voltage output and measured pressure values.

Pressure (kPa)/ Output (V)	Maximal	Typical	Minimal
0	0.4	0.2	0
5	0.8	0.6	0.4
10	1.3	1.1	0.9
15	1.7	1.5	1.3
20	2.2	2	1.8
25	2.6	2.4	2.2
30	3.1	2.9	2.7
35	3.5	3.3	3.1
40	4	3.8	3.6
45	4.4	4.2	4.0
50	4.9	4.7	4.5
55	5	4.8	4.6

The nominal transfer value: V_out_ = V_S_ (P × 0.018 + 0.04) ± (pressure error × temp. factor × 0.018 × V_S_). V_S_ = 5.0 V ± 0.25 Vdc. The temperature error factor is 1; this is a linear response from 0 °C to 85 °C according to the MPX official datasheet.

## Data Availability

The raw data supporting the conclusions of this article will be made available by the authors, without undue reservation.

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
