# Peer review of "A Custom-Tailored Multichannel Pressure Monitoring System Designed for Experimental Surgical Model of Abdominal Compartment Syndrome"

_sensors, 2024, doi:10.3390/s24020524_

Round 1

Reviewer 1 Report

Comments and Suggestions for Authors

The submission paper developed a 12-channel positive-negative pressure sensor system for simultaneous detection of pressure conditions in the abdominal cavity, intestines, and circulatory system. The system uses the piezoresistive sensors with different measurement ranges and includes analog noise filtering, A/D conversion, and microcontroller-based data processing. The system monitored pressure at different abdominal locations during negative pressure wound therapy at various suction pressures.

The following comments could be considered to improve the manuscript:

1) More details on the sensor calibration methodology could be provided. In Section 2.5, the paper describes calibrating against atmospheric pressure and taking the median of 10 measurements, but additional specifics would be helpful to understand accuracy and repeatability. Why "10 scans give the system a high degree of error tolerance"? Please interpret it.

2) Discussing any biocompatibility testing or safety evaluations of the pressure monitoring system in Section 4 would could improve the effectiveness of the proposed system. 

3) Some additional experiments validating performance over extended time periods are suggested for further robustness tests. Monitoring drift over hours-long and repeat sessions would demonstrate stability critical for chronic animal studies. Usage across multiple animals would also show reproducibility. If the authors may not be possible to include more experiments, then more discussions on reproducibility for chronic animal studies are suggested.

Author Response

Dear Reviewer,

thank you very much for your time to prepare the review, with the helpful and valuable comments. In the revised version the corrections and additions have been made, according to the comments. Below, please find the point-by-point responses.

1. According to general experience, the median value of 5 measurements ensured full error tolerance. However, there is no strict time limit for calibration. We need to read non-real-time data. The calibration should be a mean value as much as possible since all further measurement data will be shifted relative to this. A calibration reading of 10 is a good value for experience. Enough to eliminate possible air movement that touches the sensor. The number of readings can be increased at will during calibration, but we do not recommend reducing them.

2. Since the system can be connected to any sterile regular cannulas or tubes used in medicine (with appropriate connector), there is no direct issue for biocompatibility problems. We added this sentence to the text, in Section 4.

3. Further discussion and referring for experimental data have been included in Section 3 and 4.

We hope that the responses could be acceptable, and the revised version could be improved. We express our thanks again for the Reviewer’s comments, which were valuable and helpful.

Sincerely yours,

Katalin Peto, corresponding author

Reviewer 2 Report

Comments and Suggestions for Authors

In this paper (sensors-2809425), the authors designed a 12-channel positive-negative sensor system for simultaneous detection of pressure conditions in the abdominal cavity, the intestines, and the circulatory system.The strategies and research methods have a certain degree of innovation. However, there are some problems in the motivations, experimental, and data results. Some revisions need to be addressed before possible publication. My specific comments are listed below:

1.       Introduction:  What are the current difficulties in detecting abdominal compartment syndrome? It seems that sensors face more challenges (flexibility, human compatibility, stability, etc.), and it is suggested to provide relevant discussions to introduce the research strategy of this work.

2.       …three different principles: capacitive, inductive and piezoresistive [19,20].” It seems inaccurate. Compared to inductive pressure sensor, piezoelectric pressure sensors are more common (referred to J. Mater. Chem. C, 2021, 9, 13659–13667).

3.       Suggest providing sensor detection results.

4.       Is the detection system implemented? If it has already been implemented, it is recommended to provide simulated or real experimental data.

5.       Check the reference format. For example, some journal names have not been abbreviated.

6.       Check English writing.

Comments on the Quality of English Language

Minor editing of English language required.

Author Response

Dear Reviewer,

thank you very much for your time to prepare the review, with the helpful and valuable comments. In the revised version the corrections and additions have been made, according to the comments. Below, please find the point-by-point responses.

1. According to the very important suggestion, we explained the related issues and problems in the Introduction, adding these sentences: "In the clinical practice monitoring the abdominal pressure is based on pressure measurement in intra-abdominal organs such as bladder, stomach, etc. Bladder pressure - the commonly used method - correlates with intra-abdominal pressure but does not give information about the pressure in different parts of the abdomen. This question is even more interesting when negative pressure wound therapy is used for intra-abdominal hypertension or abdominal compartment syndrome. Our multi-channel system for continuous pressure measurement is sensitive to pressure changes in different parts of the abdomen. The sensors are sufficiently flexible and can be fixed with stitches or glue to well defined parts of the abdominal organs (e.g. diaphragmatic surface of the liver). With this system, the effectiveness of the inserted protective layer can be monitored not only in animal experiments but also in human practice."

2. The referred sentence has been completed adding „However, compared to inductive pressure sensors, piezoelectric pressure sensors are more common.”, and citing the paper Duan, Z.; Jiang, Y.; Huang, Q.; Yuan, Z.; Zhao, Q.; Wang, S.; Zhang, Y.; Tai, H. A do-it-yourself approach to achieving a flexible pressure sensor using daily use materials. J. Mater. Chem. C. 2021, 9, 13659-13667. doi: 10.1039/D1TC03102C.

3 and 4. Sensor detection results has been showed, as a representative graph (since it is a paper for describing the device), and we referred for the results of completed studies.

5 and 6. the format of certain references has been corrected, and English revision has been completed.

We hope that the responses could be acceptable, and the revised version could be improved. We express our thanks again for the Reviewer’s comments, which were valuable and helpful.

Sincerely yours,

Katalin Peto, corresponding author

Reviewer 3 Report

Comments and Suggestions for Authors

The authors present an interesting sensor able to test pressure relation on layers such as negative tissue wound therapy.

The electronics are presentted in detail, and the technical features are described clearly.

In my view the authors should expand on the testing of the sensor, with more details on the pressure measurements measurements presented in section three. In my view a figure that visualizes the experimental process to test the measured pressure values will help the readers understand the sensor functionality and possibilities.

I would move the BP monitoring and measurement to the discussion and address possible implementation issues along with some references. 

Comments on the Quality of English Language

English is OK.

Author Response

Dear Reviewer,

thank you very much for your time to prepare the review, with the helpful and valuable comments. In the revised version the corrections and additions have been made, according to the comments: section 3 has been extended and completed, and the idea for BP monitoring has been moved to the Discussion part.

We hope that the responses could be acceptable, and the revised version could be improved. We express our thanks again for the Reviewer’s comments, which were valuable and helpful.

Sincerely yours,

Katalin Peto, corresponding author

Round 2

Reviewer 2 Report

Comments and Suggestions for Authors

The response and revised manuscript are satisfactory, and it is recommended to accept.

Reviewer 3 Report

Comments and Suggestions for Authors

The authors have addressed the revisions successfully.

Comments on the Quality of English Language

English is  ok